# *Ficus dubia* Latex Extract Induces Cell Cycle Arrest and Apoptosis by Regulating the NF-κB Pathway in Inflammatory Human Colorectal Cancer Cell Lines

**DOI:** 10.3390/cancers14112665

**Published:** 2022-05-27

**Authors:** Rentong Hu, Weerachai Chantana, Pornsiri Pitchakarn, Subhawat Subhawa, Bhanumas Chantarasuwan, Piya Temviriyanukul, Teera Chewonarin

**Affiliations:** 1Department of Biochemistry, Faculty of Medicine, Chiang Mai University, 110 Intravaroros Rd., Sripoom, Muang, Chiang Mai 50200, Thailand; hurentong@163.com (R.H.); james301239@gmail.com (W.C.); pornsiri.p@cmu.ac.th (P.P.); 2Department of Laboratory Medicine, The Affiliated Hospital of Youjiang Medical University for Nationalities, Baise 533000, China; 3Clinical Research Center for Food and Herbal Product Trials and Development (CR-FAH), Faculty of Medicine, Chiang Mai University, Chiang Mai 50200, Thailand; subhawat.s@cmu.ac.th; 4Thailand Natural History Museum, National Science Museum, Pathum Thani 12120, Thailand; bhanumas@nsm.or.th; 5Food and Nutrition Academic and Research Cluster, Institute of Nutrition, Mahidol University, Salaya, Phuttamonthon, Nakhon Pathom 73170, Thailand; piya.tem@mahidol.ac.th

**Keywords:** colorectal cancer, *Ficus dubia* latex extract (FDLE), inflammation, proliferation, cell cycle arrest, apoptosis, NF-κB pathway

## Abstract

**Simple Summary:**

In this study, the anti-tumor activity of *Ficus dubia* latex extract (FDLE) against HCT-116 and HT-29 human colorectal cancer cell lines in normal and inflammatory conditions was investigated. The results indicated that an inflammation-activated NF-κB increased proliferation-related proteins and decreased apoptosis-related proteins, leading to hyperproliferation of colorectal cancer cell lines. FDLE exhibited remarkable antiproliferative activity in both conditions, however, more effective anti-proliferation was observed due to its more effective regulation of NF-κB inactivation and some proteins related to cell cycle progression and apoptosis induction in inflammatory condition. Our finding may promote the antiproliferative study of FDLE in inflammation-induced colorectal cancer progression.

**Abstract:**

Colorectal cancer is one of the most diagnosed cancers that is associated with inflammation. *Ficus dubia* latex is recognized as a remedy with various therapeutic effects in traditional medicine, including anti-inflammatory and antioxidant activity. The present study aims to compare the anti-tumor activity of *Ficus dubia* latex extract (FDLE) against HCT-116 and HT-29 human colorectal cancer cell lines in normal and inflammatory condition and explore its mechanism of action. FDLE exhibited remarkable antiproliferative activity against HCT-116 and HT-29 colorectal cancer cell lines in both conditions using MTT and colony formation assays and more effective anti-proliferation was observed in inflammatory condition. Mechanistically, FDLE induced cell cycle arrest at G0/G1 phase by down-regulating NF-κB, cyclin D1, CDK4 and up-regulatingp21 in both cell in normal condition. In inflammatory condition, FDLE not only exhibited stronger induction of cell cycle arrest in both cells by down-regulating NF-κB, cyclin D1, CDK4 and down-regulating p21, but also selectively induced apoptosis in HCT-116 cells by down-regulating NF-κB and Bcl-xl and up-regulating Bid, Bak, cleaved caspase-7 and caspase-3 through stronger ability to regulate these proteins. Our results demonstrated that the phytochemical agent in the latex of *Ficus dubia* could potential be used for treatment and prevention of human colorectal cancer, especially in inflammation-induced hyperproliferation progression.

## 1. Introduction

Colorectal cancer (CRC) is among the most prevalent malignant tumors and the second largest cause of cancer-related mortality [1,2]. Inflammatory microenvironment is thought to be responsible for the development and progression of CRC [3]. In addition, patients with CRC who have inflammatory responses appear to report a higher rate of tumor recurrence and a greater number of adverse effects from therapeutic and surgery treatments than those without inflammation [4]. In the inflammation process, pro-inflammatory cytokines such as tumor necrosis factor–α (TNF-α) and interleukin-1β (IL-1β) consequently activate transcriptional regulatory signaling pathways and cooperate to regulate cancer-related pathophysiological processes including proliferation [5]. Similarly, lipopolysaccharide (LPS) is a substance that induces the inflammation process and can be promoted by TNF-α and Interferon-γ (IFN-γ) in colorectal cancers [6]. Consequently, the inflammatory model induced by LPS combined with TNF-α and IFN-γ has been used to investigate the intervention effects of target compounds against inflammatory colorectal cancer [7,8].

Nuclear factor-κB (NF-κB) is an important inflammatory signaling pathway and its mediated transcription plays a critical role in cancer development. The activated NF-κB translocates into the nucleus and mediates the expression of target genes associated with cell proliferation, apoptosis and inflammation [9,10]. On the one hand, NF-κB promotes cell proliferation by regulating cell cycle-related proteins [11]. On the other hand, NF-κB is regarded as a major antiapoptotic factor, since it not only activates the expression of encode anti-apoptotic proteins such as inhibitor of apoptosis protein (IAP), B-cell lymphoma 2 (Bcl-2) and B-cell lymphoma-extra-large (Bcl-xl), but also inactivates the expression of apoptotic proteins including BH3 interacting-domain death agonist (Bid), Bcl-2 associated x (Bax) and Bcl-2 associated k (Bak), then activates caspase cascade enzymes such as cleaved caspase-7 and caspase-3, which directly induce apoptosis [12]. To prevent colorectal tumors, inhibition of these pathways is an effective strategy. 

Chemotherapy and radiotherapy are common colorectal cancer treatments, but their side effects, such as nausea and diarrhea, impair the quality of life [13]. Therefore, to relieve the symptoms of pain and to have better overall health, more effective therapeutic strategies are requisite in the treatment of CRC. In the past few years, a wide number of herbs and/or herbal extracts with anti-oxidant, anti-inflammation and anti-proliferation properties, such as green tea extract, apple extract, soy isoflavones and triptolide are being frequently consumed by both healthy people and patients of CRC [14,15]. Therefore, numerous efforts have been dedicated to the invention of natural anticancer medications, and natural functional herbs are of interest as possible anticancer drugs [16]. 

*Ficus dubia* as a new *Ficus* species is an indigenous plant that is abundantly found in tropical Asia, including the southern part of Thailand, Malaysia, Indonesia and Brunei [17]. It is used in traditional drugs due to belief in its health benefits. In fact, Suttisansanee et al. [18] found that the extract of the *Ficus dubia* sap was rich in flavonoids and phenolics, the extract exhibited antioxidant activities in in vitro experiments and also suppressed lung and ovarian cancer growth by inducing cell cycle arrest at the G1 and G2/M phase. In addition, Chansriniyom et al. [19] found that the extract of *Ficus* dubia sap exhibited antioxidant activity against H_2_O_2_-induced oxidative stress in HaCaT cells. Moreover, Rodthayoy et al. found that the extract of *Ficus dubia* latex, another rich source of flavonoids and phenolics, exhibited anti-inflammatory activity in lipopolysaccharide-induced macrophages [20]. Interestingly, data to support the anti-tumor biological activities of other *Ficus* species has been reported in several studies. For example, Abolmaesoomi et al. [21] found that the extract of *Ficus deltoiea* exhibited anti-proliferative activities in HCT-116 human colon cancer cells. In addition, *Ficus carica* latex extract was also found to have anti-proliferation activity against colorectal cancer cell lines [22]. Our previous research also found that the extract of *Ficus* dubia latex inhibited DMH-induced rat colorectal carcinogenesis by investigating xenobiotic metabolism, inflammation, proliferation and apoptosis [23]. However, the antiproliferative mechanism is not precisely understood, especially in inflammatory states. Therefore, the present study aims to compare the anti-tumor activity of *Ficus dubia* latex extract (FDLE) against HCT-116 and HT-29 human colorectal cancer cell lines in normal and inflammatory condition and explore its mechanism of action.

## 2. Materials and Methods

### 2.1. Chemicals and Reagents

*Ficus dubia* (voucher number: Chantarasuwan 040117-1) used in this study was submitted by Bhanumas Chantarasuwan in the National Science Museum, Thailand and the water extraction of latex was conducted and provided by the Institute of Nutrition, Mahidol University, as described in a previous study [19]. DMEM medium and FBS were purchased from GIBCO (New York, NY, USA); Antibodies of NF-κB, p-NF-κB, PCNA and β-actin were obtained from Cell Signaling (Beverly, MA, USA). Antibodies of cyclin D_1_, CDK4, p21, Bid, Bak, Bcl-xl, caspase-7 and caspase-3 were obtained from ABCAM (Waltham, MA, USA); goat anti-mouse IgG, goat anti-rabbit IgG and Annexin V-FITC Apop Kit were purchased from Thermo Fisher Scientific (Waltham, MA, USA). Other reagent grade chemicals were obtained from Merck Millipore Bioscience (Bangkok, Thailand).

### 2.2. Cell Lines and Culture Condition

Human colorectal cancer cell lines (HCT-116 and HT-29) and normal mouse fibroblasts (NIH3T3) were purchased from ATCC (American Type Culture Collection, Manassas, VA, USA). These cell lines were cultured in Dulbecco’s Modified Eagle Medium (DMEM) with 10% fetal bovine serum (FBS) and 1% Penicillin-Streptomycin at 37 °C in a humidified atmosphere containing 5% CO_2_. 

### 2.3. The Experimental Design of Non-Inflammatory and Inflammatory Model

HCT-116, HT-29 and NIH3T3 cells were cultured and incubated at 37 °C and 5% CO_2_ overnight. For the inflammation model, each cell line was cultured in DMEM with a mixture of cytokines (TNF-α, IFN-γ and LPS each 10 ng/mL) for two hours before FDLE treatment, whereas a vehicle was added instead of cytokine mixed in non-inflamed condition. Then, inflamed and non-inflamed cells were added with various concentrations of FDLE (0, 25, 50, 100, 150 and 200 µg/mL) for 24, 48 and 72 h to MTT, and various concentrations of FDLE (50, 100, and 200 µg/mL) for 48 h to colony formation assay, cell cycle arrest, apoptosis induction and Western blot analysis. The experimental design of the non-inflammatory and inflammatory models is shown in Figure 1.

### 2.4. Measurement of Relative Cell Viability Using MTT Assay

Treated HCT-116, HT-29 and NIH3T3 cells (1 × 10^4^ cells) in 96-well plate as described in 2.3 were added to 15 µL of 5 mg/mL MTT dye (3-(4,5-dimethylthiazol-2-yl)-2,5-diphenyltetrazolium bromide). The incubation was carried out at 37 °C for four hours. All solution was discarded and 100 µL DMSO was added to dissolve formazan crystal. The solution was subjected to measure an absorbance at 540/630 nm using a microplate reader. A percentage of cell viability was calculated against untreated cells in each time period. The experiments were repeated three times with at least triplicate per experiment, the values of CI50 and CI20 were analyzed with GraphPad software. 

### 2.5. Determination of Colony Formation by Colony Formation Assay 

Treated HCT-116 and HT-29 cells (500 cells) in 6-well plate as described in 2.3 were washed two times with phosphate buffer saline (PBS) and then incubated at 37 °C for seven days. The surviving cells could proliferate and form a colony during this period. On day 7, the colonies were washed with PBS and fixed by adding absolute methanol for 5–10 min. After that, the fixed cells were stained with 0.5% crystal violet for at least one hour. The visualized colonies were counted and compared to untreated cells, which were expressed as colony formation rates (%). The experiments were performed as three independent experiments. 

### 2.6. Analysis of Cell Cycle Arrest by Flow Cytometry

HCT-116 and HT-29 cells (1 × 10^5^ cells) in 6-well plate were synchronized into G0/G1 phase by serum starvation for 24 h after being cultured overnight. Then, starved cells were washed two times with PBS and treated as described in 2.3. The treated cells were collected and washed with cold PBS three times. Cells were fixed in cold 70% ethanol for 18 h. After washing, the fixed cells were mixed with 100 µg/mL of ribonuclease A and 50 µg/mL of propidium iodine (PI) for 30 min on ice. After incubation, labeled cells were washed with cold PBS twice and subjected to flow cytometry. The intensity of fluorescence was analyzed by FlowJo Version 7.6.1 (FlowJo LLC, Ashland, OR, USA) to identify cell cycle arrest stages. The experiments were performed as three independent experiments. 

### 2.7. Analysis of Apoptosis Induction by Flow Cytometry

Treated HCT-116 and HT-29 cells (1 × 10^5^ cells) in 6-well plate as described in 2.3 were collected and cell pellets were washed with cold PBS three times. Each pellet was suspended in 400 µL of cold binding buffer cell suspension, then mixed with 3 µL of annexin V and 2 µL of propidium iodide (PI) (Merck, Darmstadt, Germany); the mixture was further incubated on ice for 15 min in the dark. Labeled cells were subjected to flow cytometry. The intensity of fluorescence between fluorescent Annexin V-FITC and PI were analyzed by FlowJo Version7.6.1 (FlowJo LLC, Ashland, OR, USA) to identify apoptotic stages. Apoptosis induction of FDLE was evaluated for the percentage of early and late apoptotic cells compared to untreated cells (control). The experiments were performed as three independent experiments. 

### 2.8. Measurement of Proteins Related to Cell Cycle Arrest and Apoptosis by Western Blot 

Total protein from conditioned HCT-116 and HT-29 cells homogenate as described in 2.3 was collected and separated by SDS-polyacrylamide gel electrophoresis (PAGE) and the separated proteins were then transferred onto nitrocellulose membrane. The transferred proteins were detected using specific primary anti-NF-κB, anti-p-NF-κB, anti-cyclin D1, anti-CDK4, anti-P21, anti-PCNA, anti-Bid, anti-Bak, anti-Bcl-xl, anti-caspase-7, anti-caspase-3 (1:3000) and anti-β-actin (1:5000) overnight at 4 °C and secondary anti-rabbit or anti-mouse (1:5000) were labeled with horse radish peroxidase enzyme followed by chemiluminescent substrate. The specific target proteins were visualized by adding western lightening chemiluminescent HRP Substrate (PerkinElmer, Waltham, MA, USA), and the pictures were captured by Biorad ChemiDoc XRS System (Bio-Rad, Hercules, CA, USA). Intensity of target proteins were measured by Image J program. The band intensity of each protein was normalized against β-actin and the relative expression in each condition was compared to the non-treatment group. 

### 2.9. Statistical Analysis

The measurable data were calculated and the mean ± SD was expressed. The significant difference between the mean of the groups was analyzed by one way analysis of variance (ANOVA). The results were considered statistically significant at * *p* < 0.05, ** *p* < 0.01, *** *p* < 0.001, **** *p* < 0.0001. All the image quantifications were done by Image-Pro (version 7.0). All the statistics were performed on GraphPad Prism 6.0 (GraphPad Software, San Diego, CA, USA).

## 3. Results

### 3.1. Subsection Effect of FDLE on Cell Viability in Colorectal Cancer Cell Lines and Normal Mouse Embryonic Fibroblast Cell (NIH3T3) by MTT Assay

The percentages of cell viability of each cell lines at indicated times after treatment with FDLE both with and without inflammatory conditions are shown in Figure 2. Administration of FDLE higher than 100 µg/mL had a significant effect on both HCT-116 and HT-29 cell lines in a dose dependent manner when treated for 48 and 72 h (Figure 2A,B). On the other hand, FDLE did not affect the growth of NIH3T3, normal cell line (Figure 2C). TNF-α, IFN-γ and LPS each 10 ng/mL significantly increased cell viability in the three cells when compared to non-treated cells. FDLE administration not only exhibited substantial cell viability inhibition in HCT-116 and HT-29 cells, but also in NIH3T3 cells, when treated for 48 and 72 h, excluding 24 h in a dose-dependent manner (Figure 2D–F). In addition, the IC_20_ and IC_50_ values of FDLE for both HCT-116 and HCT-29 cell lines in inflammatory condition were lower than in normal condition when treated for 48 and 72 h, respectively (Appendix A), which showed that FDLE was more effective in inflammatory state. Since FDLE started to exhibit the inhibitory effect after being treated with 100 µg/mL FDLE for 48 h, various FDLE (50, 100 and 200 µg/mL) were used in all further experiments for exploring the anti-proliferative effect and understanding molecular mechanisms leading to growth arrest and apoptosis after treatment with FDLE for 48 h.

### 3.2. Effects of FDLE on Growth Inhibition in Colorectal Cancer Cell Lines by Colony Formation Assay

According to the results of MTT assay, FDLE tended to reduce proliferation in HCT-116 and HT-29 colorectal cancer cell lines in both normal and inflammatory conditions. Therefore, the inhibitory effect of FDLE on seven days’ growth of these cell lines was examined. The colony number of HCT-116 and HT-29 are shown in Figure 3. FDLE significantly reduced the colony number of HCT-116 and HT-29 cells in normal condition in a dose-dependent manner, when compared to untreated cells (Figure 3A–C). In inflammatory condition, mixture of TNF-α, IFN-γ and LPS significantly increased colony number of both HCT-116 and HT-29 cells, while in the FDLE treatment, the colony formations were both significantly reduced in a dose-dependent manner, when compared to cytokines-treated cells (Figure 3D–F). The growth inhibition rates of 200 µg/mL FDLE on HCT-116 cells in normal and inflammatory conditions were 47.8% and 58.4%, and HT-29 cells were 46.5% and 56.9%, respectively (Appendix A). The results indicated that FDLE revealed inhibitory effects against normal growth colorectal cancer cells together with the growth under inflammatory conditions, whereas the inhibitory effect was more observable in inflammatory condition.

### 3.3. Effects of FDLE on Cell Cycle Arrest in Colorectal Cancer Cell Lines by Flow Cytometer

According to anti-proliferation of colorectal cancer cells by FDLE, the modulatory effect on the cell cycle was interesting to determine. The distribution of the cell population in each phase of the cell cycle at 48 h was measured and analyzed in each condition compared to positive control BEZ235 (200 ng/mL), as shown in Figure 4. In normal growth, FDLE at doses of 100 and 200 µg/mL significantly decreased S and G2/M phase together with increasing the population of G0/G1 phase in both HCT-116 and HT-29 cells, respectively, when compared to untreated cells (Figure 4A–C). Interestingly, mixture of cytokines significantly increased S and G2/M phase in both HCT-116 and HT-29 cells, when compared to untreated cells, which was related to the enhancement of cancer growth rate. FDLE administration at doses of 50, 100 and 200 µg/mL all significantly decreased S and G2/M phases in both HCT-116 and HT-29 cells, when compared to inflamed cells (Figure 4D–F). Meanwhile, FDLE increased the ratio of cell cycle arrest at the G0/G1 phase, leading to the reduction of cancer cells growth. The effect of FDLE on G0/G1 arresting was more clearly observed in inflammatory condition than in the normal growth of both HCT-116 and HT-29 cells (Appendix A).

### 3.4. Effect of FDLE on NF-κB-Mediated Proteins Related to Cell Cycle by Western Blot

To investigate the molecular mechanism of FDLE on cell growth inhibition, proteins related to cell cycle including cyclin D1, CDK4, p21, PCNA and inflammatory signaling mediator NF-κB (p65) were determined by immunoblot and the intensity of protein bands is shown in Figure 5. In normal condition, FDLE administration significantly reduced p-NF-κB (p65), cyclin D1, CDK4 and increased p21, but had no effect on PCNA in both HCT-116 and HT-29 cells, when compared to untreated cells (Figure 5A,B). In an inflammatory condition, a mixture of cytokines strongly increased p-NF-κB (p65), cyclin D1 and CDK4, while significantly decreasing p21 in HCT-116 cells. However, treatment with a mixture of cytokines had no apparent effect on the expression of p21 on HT-29 cells, while increasing p-NF-ΚB (p65), cyclin D1 and CDK4 when compared to untreated cells. Interestingly, after FDLE treatment for 48 h, the levels of p-NF-κB (p65), cyclin D1 and CDK4 were decreased, but p21 was increased when compared to cytokines-treated cells alone in both HCT-116 and HT-29 cells (Figure 5C,D). These results demonstrated that FDLE induced cell cycle arrest via inactivation of NF-κB signaling pathways resulting in the downregulation of cyclin D1 and CDK4 expression, whereas p21 upregulation in both normal and inflammatory condition. In the same way, FDLE was more effective to regulate the inactivation of NF-ΚB and expression of protein related cell cycle progression during the inflammation-induced cancer cell hyperproliferation process (Appendix A).

### 3.5. Effects of FDLE on Apoptosis Induction of Colorectal Cancer Cell Lines by Flow Cytometry

To explore whether the anti-proliferative activity leading to apoptosis, Annexin V-FITC/PI staining of treated cells was performed and apoptotic cells were determined by flow cytometry. In the normal growth of cancer cells, FDLE administration did not induce apoptosis in both HCT-116 and HT-29 cells when compared to untreated cells (Figure 6A–C). Interestingly, the mixture of cytokines significantly reduced the number of apoptotic cells only in HT-116 cells, but not in HT-29 cells. FDLE administration was found to significantly induce apoptosis in inflamed HCT-116 cells in a dose-dependent manner, when compared to cytokines-treated cells; however, the apoptosis in inflamed HT-29 cells treated with FDLE was not clearly observed (Figure 6D–F). Our results indicated that FDLE did not induce apoptosis in non-inflamed HCT-116 and HT-29, whereas it was able to induce apoptosis only in HCT-116 treated by a mixture of cytokine, although this inhibitory effect was not observed in inflamed HT-29 cells in the same condition. In inflammatory conditions, the apoptosis ratio of HCT-116 was more significant than that of normal growth, while HT-29 did not differ between the two conditions (Appendix A). These results further suggested that FDLE could increase apoptosis only in the hyper-proliferated colorectal cancer via anti-inflammatory action.

### 3.6. The Effect of FDLE on NF-ΚB Mediated Expression of Apoptotic Proteins and Activation of Caspase Enzymes by Western Blot

The molecular mechanism of FDLE on apoptosis induction was also further investigated. In normal condition, FDLE administration only significantly reduced p-NF-κB (the level of p65), but had no effect on the expression of Bid, Bak and Bcl-xl, nor on the cleaved caspase-7, and caspase-3 in both HCT-116 cells (Figure 7A) and HT-29 cells (Figure 7B). These results revealed that FDLE did not significantly induce apoptosis in either of the colorectal cancer cell lines. According to the inflammation, TNF-α, IFN-γ and LPS could induce cell proliferation via NF-κB signaling pathways [6,8]. A mixture of cytokines significantly increased p-NF-κB and decreased Bid, Bak, cleaved caspase-7, and caspase-3, except for Bcl-xl, which was not changed in HCT-116 cells. For HT-29 treated with inflammation conditions, there was an increase in the p-NF-κB and a decrease in Bid and Bak. Noticeably, FDLE administration induced apoptosis in HCT-116 cells according to significantly decreased p-NF-κB and Bcl-xl, and increased Bid and Bak, resulting in an increasing of cleaved caspase-7 and caspase-3, when compared to cytokines-treated group (Figure 7C,D), which was consistent with the flow cytometric analysis. Nonetheless, the effect of FDLE under inflammatory conditions was not substantially demonstrated in HT-29, where FDLE only inhibited p-NF-κB and increased in Bid and Bak (Figure 7D). As a result, it was found that only FDLE was insufficient to induce apoptosis in colorectal cancer cells. However, in inflammatory conditions, it can promote apoptosis, and the effect was more significant in HCT-116 cells than in HT-29 cells due to an inhibition in the NF-κB signaling pathway. FDLE was also more effective to regulate the inactivation of NF-ΚB and expression of protein-related apoptosis during the inflammation-suppressed cancer cell apoptosis process (Appendix A).

## 4. Discussion

*Ficus dubia* has been identified as a potent anti-oxidant and anti-inflammatory activity because of its high phytochemical contents [19,20]. In a previous study, we discovered that the extract of *Ficus dubia* latex includes 34.2% phenolic and 7.3% flavonoids; specifically, the amount of chlorogenic acid, one form of phenolic, is approximately 2.1% using HPLC compared to the chlorogenic acid standard, with an extraction yield of 7.67% [19]. Furthermore, there was a report about the compounds found in FDLE, including quinic acid and caffeoyl derivatives (syringoylquinic acid, 3-O-caffeoylquinic acid, 4-O-caffeoylquinic acid, and dimeric forms of caffeoylquinic acids) [19], which was consistent with our previous study [23]. Chlorogenic acid and its metabolites have been found to exert anti-proliferative effects by inducing S-phase cell cycle arrest and apoptosis in human colon cancer caco-2 cells [24]. Despite the fact that the stability of FDLE has not been investigated, the pharmacokinetics of chlorogenic acids, the major component of FDLE, has been studied by various of researchers in humans and animals [25]. Therefore, chlorogenic acid is an effective active ingredient that have been use as standard content for evaluating the quality of extract preparation by HPLC.

Our previous study (in press) also demonstrated that FDLE revealed inhibition in DMH-induced rat colorectal carcinogenesis and anti-inflammation in LPS-induced RAW 264.7 cells [23]. However, the anti-proliferative molecular mechanism was not precisely understood. In this regard, the inhibitory effect of FDLE on HCT-116 and HT-29 colorectal cancer cell lines and the possible molecular mechanism were explicated. Firstly, TNF-α caused inflammation in cancers was commonly implemented in numerous investigations [26]. However, there is no evidence for the use of TNF- α and LPS to induce cell proliferation and inflammation in cancer cells. According to a previous study, LPS could affect the expression of inflammatory factors such as TNF-α in the SW480 and HCT116 colorectal cancer cell lines via NF-κB signaling pathway but had no effect on cell proliferation [6,27]. However, our MTT assay exhibited that a combination of TNF-α, IFN-γ and LPS could induce cell proliferation (Figure 2D–F) and colony forming assay (Figure 3D–F), which is an inflammation in the cancer microenvironment through the activation of NF-κB [28] (Figure 5C,D). Based on our findings with HCT-116 and HT-29 cells in the MTT assay and colony forming assay, FDLE has the potential to decrease cell proliferation under both normal and inflammatory conditions. In addition, it was found that at an IC_50_ of FDLE in colorectal cancer cells greater than 200 µg/mL, it was not cytotoxic to inflamed NIH3T3 cells (Appendix A). Interestingly, more effective in anti-proliferation was clearly observed in both MTT and colony forming assay, resulting in lower IC_20_, IC_50_ values and greater inhibition of colony formation in inflammation environment, which may be due to the higher sensitivity of FDLE on inflamed colorectal cancer cells (Appendix A). Consequently, the results primarily suggested that FDLE retarded the growth rate of inflamed cancer cell which may be due to its anti-inflammatory activity. The reports of anti-proliferation by interruption of the inflammatory process have supported our findings.

To explain the effects of FDLE at the cellular level, cell cycle and its regulatory proteins was used, a biomarker for proliferation and attractive therapeutic targets for development of anticancer drugs have been determined [29]. Our results showed that inflammation activated NF-κB, unregulated cyclin D1 and CDK4 and downregulated p21 leading to the hyperproliferation by promoting cell cycle progression. FDLE arrested cancer cells at the G0/G1 phase by downregulating p-NF-κB, cyclin D1 and CDK4, while upregulating p21 in both normal growth and inflamed HCT-116 and HT-29 cells. The reduction of CDK4 and cyclin D was able to break the cell cycle progression from G0 to early G1 phase whereas the higher level of p21 can additionally inhibit the cyclin D/CDK complex [30]. The previous study has supported that the extracts of *Ficus carica* leaves has been shown to inhibit cell proliferation by inducing cell cycle arrest in triple-negative breast cancer MDA-MB-231 cells [31]. Our results suggested that FDLE-induced cell cycle arrest by the inactivation of NF-κB signaling, because the effect of FDLE on the ratio of G0/G1 arresting was clearly observed in inflammation conditions than in the normal growth of both HCT-116 and HT-29 cells (Appendix A). To test our hypothesis, we further compared the ability of FDLE to regulate NF-κB and its mediated cell proliferation-related proteins in different conditions. We found that FDLE at dose of 200 µg/mL exhibit higher relative change ratio to p-NF-κB, cyclin D1, CDK4 and p21 proteins in inflammatory condition when compared to non-inflammatory condition (Appendix A), which indicated that stronger regulatory ability under inflammatory conditions may be an important reason for more effective anti-proliferative effect, especially, the inactivation of NF-κB. Several reports have shown that the inactivation of NF-κB can suppress cancer cell growth. For example, *Inonotus obliquus* suppressed proliferation of colorectal cancer cells and tumor growth in mice models by down-regulating NF-κB signaling pathways [32]. Therefore, the anti-inflammatory activity may be accountable for the more effective anti-proliferative potential via inhibition of NF-κB signaling pathway shown by FDLE in the inflammatory environment.

Apoptosis induction is also considered an important strategy to inhibit tumor cell proliferation [33]. Due to FDLE’s low toxicity, it was not able to induce apoptosis of HCT-116 and HT-29 cells. Surprisingly, only cytokine-activated HCT-116, but not HT-29 underwent apoptosis when treated with FDLE. Zhang et al. [34] found that inflammation increases the cytotoxicity of isoniazid leading to higher apoptosis induction in LPS-induced zebrafish. Similarly, Hillyer and Krishnan also found that vitamin D exhibits stronger antiproliferative effects in inflammatory breast cancer cell lines by inducing apoptosis, due to its anti-inflammatory activity [35,36].

In non-inflammation of HCT-116, FDLE only decreased the expression of the anti-apoptotic protein Bcl-xl and slightly increased cleaved caspase-7, but the effect was not observed in HT-29. Interestingly, in inflamed HCT-116 cells, FDLE down-regulated Bcl-xl and up-regulated Bid, Bak, cleaved caspases-7 and caspase-3, leading to activate apoptosis process. These observations were related to the inactivation of NF-κB by FDLE. However, in inflamed HT-29 cells, although FDLE inactivated NF-κB and upregulated Bid and Bak, it does not activate caspase cascade enzymes. Therefore, caspase cascade activation is a key step of apoptosis induction and the NF-κB signaling pathways is an upstream target for inhibiting colorectal cancer, especially in the inflammatory condition. Similarly, Yousef et al. [37] found that Pristimerin exhibited apoptosis induction through inhibition of NF-κB signaling pathways in HCT-116 colorectal cancer cells induced by TNF-α and LPS. Hillyer and Krishnan also found that more effective apoptosis of Vitamin D in inflammatory breast cancer cells by modulation of NF-κB activation regulated Bcl-2 family and activated caspase cascades to induce apoptosis [35,36].

We then tried to explain why FDLE had more sensitivity to apoptosis induction by FDLE in inflamed HCT-116 cells than in inflamed HT-29 cells. Similarly, HCT-116 cells were shown to have a significant higher sensitivity to saffron extract compared to other colorectal cancer cell lines SW-480 and HT-29 [38]. Baldari’s study also found that copper chelation had a stronger sensitivity to HCT-116 cells, compared to HT-29 cells [39]. At the molecular level, the possible reason for this was these two colorectal cancers cells used in the study had different mutations status in several critical genes involved in colorectal cancer in addition to BRAF (HCT-116 wt; HT-29 V600E) including P53 (HCT-116 wt; HT-29 mutation R273H), K-ras (HCT-116 mutation G13D; HT-29 wt) and PIK3CA (HCT-116 H1047R; HT-29 wt) [40]. Gene mutations in APC, K-ras, and p53 were thought to be essential events for colorectal cancer development, in particular, K-ras and p53 mutations rarely co-exist in the same tumor [41]. The mutation of K-ras, and p53 genes leading to the activation of K-ras oncoprotein and the inactivation of p53 tumor suppressor protein, have been implicated in drug resistance [42,43]. Bajbouj et al. [44] found that Saffron extract had stronger apoptosis induction in HCT-116 p53 wild type cells than in HCT-116 p53 −/− cells. Arisan et al. [45] also found that induced apoptosis of diclofenac in HCT 116 was more efficient, when compared to SW480 colorectal cancer cells, since HCT-116 was p53 wild type and WS480 was p53R273H mutant. According to p53, p21 in tumor cell response to cytotoxic medicines, as well as the potential to increase the therapeutic index of such agents by altering p21 status through p53-dependent pathways, which was detected in HCT-116 [46], but not in HT29, which has a deficiency in p53 gene expression [47]. As a result, HCT-116 was significantly more sensitive to FDLE than HT-29.

Considering normal and inflammatory conditions in both colorectal cancer cell lines, we also compared the ability of FDLE to regulate NF-κB and its mediated apoptosis-related proteins in different conditions. Relative higher change ratio of NF-κB inactivation and caspase cascade activation may be responsible for higher apoptosis induction in inflammatory condition when compared to non-inflammatory condition (Appendix A). Overall, FDLE was able to strongly suppress cancer cell growth in HCT-116 cells by decreasing growth-related proteins such as cyclin D1 and CDK4 and increasing p21 through the mechanism of inhibition of NF-kB signaling pathway. Similarly, FDLE was significantly reduced in HT-29 cells, but not as much as in HCT-116. According to inducing apoptosis, FDLE was only able to induce apoptosis cell death in HCT-116 cell under inflammatory conditions; however, in other situations or even other cells, FDLE is not toxic enough to induce apoptosis. Consequently, it is evident that FDLE has a higher inhibitory effect in inflammatory situations than apoptosis induction, except in inflammatory conditions involving HCT-116. We believe that FDLE’s anti-tumor actions on colorectal cancer cells may be investigated further in clinical trials based on the findings of this study.

## 5. Conclusions

Our studies demonstrated that an inflammation-activated NF-κB increased proliferation-associated proteins and decreased apoptosis-associated-proteins leading to hyperproliferation of colorectal cancer cell lines. The latex of *Ficus dubia* exhibited anti-proliferative activity in both normal growth and inflamed HCT-116 and HT-29 cells, and this anti-proliferative activity was more effective in the inflammatory condition. In non-inflammatory condition, FDLE only induced cell cycle arrest by down-regulating NF-κB, cyclin D1, CDK4 and increased p21 in both HCT-116 and HT-29 cells. In inflammatory condition, FDLE not only exhibited stronger induction of cell cycle arrest in both cells by down-regulating NF-κB, cyclin D1, CDK4 and down-regulating p21, but also selectively induced apoptosis in HCT-116 cells by down-regulating NF-κB and Bcl-xl and up-regulating Bid, Bak, cleaved caspase-7 and caspase-3 through a stronger ability to regulate these proteins. Therefore, the phytochemical agent in latex of *Ficus dubia* could potential be used for treatment and prevention of human colorectal cancer, especially in inflammation-induced hyperproliferation progression. The proposed anti-proliferative mechanisms of FDLE on human colorectal cancer cell lines is shown in Figure 8.

## Figures and Tables

**Figure 1 cancers-14-02665-f001:**
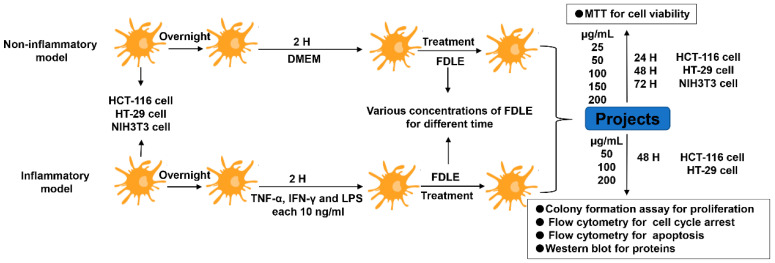
The experimental design of non-inflammatory and inflammatory models.

**Figure 2 cancers-14-02665-f002:**
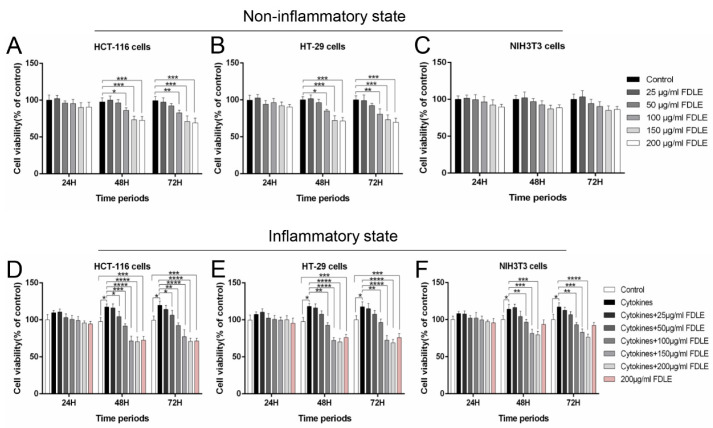
The effects of FDLE on cell viability of colorectal cancer cell lines (HCT-116 and HT-29) and normal mouse embryonic fibroblast cell (NIH3T3) in non-inflammatory and inflammatory states by MTT. The cells were treated with various concentrations of FDLE for 24, 48 and 72 h. The cell viability was calculated comparing to untreated control cells after 24, 48 and 72 h of incubation. (**A**–**C**) The cell viability of HCT-116, HT-29 and NIH3T3 cells in non-inflammatory state. (**D**–**F**) The cell viability of HCT-116, HT-29 and NIH3T3 cells in inflammatory state. The results were represented as mean ± SD of three independent experiments, * *p* < 0.05, ** *p* < 0.01, *** *p* < 0.001, **** *p* < 0.0001.

**Figure 3 cancers-14-02665-f003:**
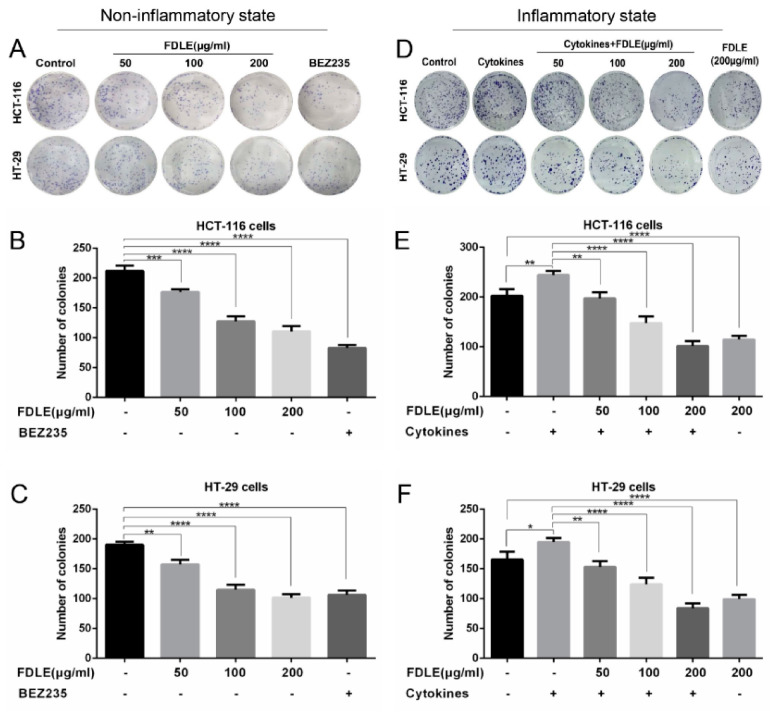
The effects of FDLE on growth inhibition in non-inflammatory and inflammatory HCT-116 and HT-29 colorectal cancer cells by colony formation assay. (**A**–**C**) Colony number of non-inflammatory HCT-116 and HT-29 cells treated with three doses (50, 100 and 200 µg/mL) of FDLE for 48 h. (**D**–**F**) Colony number of inflammatory HCT-116 and HT-29 cells treated with three doses (50, 100 and 200 µg/mL) of FDLE for 48 h after adding a mixture of cytokines (TNF-α, IFN-γ and LPS each 10 ng/mL) for 2 h. The cells were washed by PBS and cultured for 7 days. The results were represented as mean ± SD of three independent experiments, * *p* < 0.05, ** *p* < 0.01, *** *p* < 0.001, **** *p* < 0.0001.

**Figure 4 cancers-14-02665-f004:**
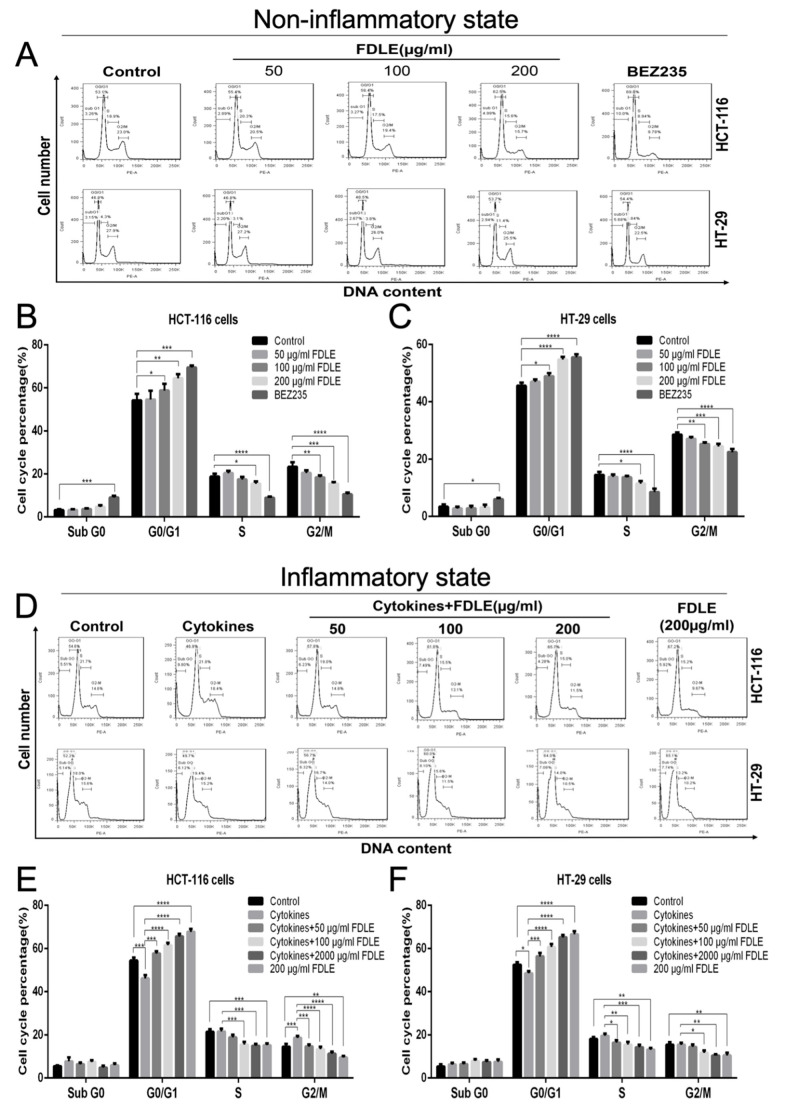
The effects of FDLE on cell cycle arrest in non-inflammatory and inflammatory HCT-116 and HT-29 colorectal cancer cells by flow cytometry. (**A**–**C**) Representative cell cycle distribution of non-inflammatory HCT-116 and HT-29 cells treated with three doses (50, 100 and 200 µg/mL) of FDLE for 48 h. (**D**–**F**) Representative cell cycle distribution of inflammatory HCT-116 and HT-29 cells treated with three doses (50, 100 and 200 µg/mL) of FDLE for 48 h after adding a mixture of cytokines (TNF-α, IFN-γ and LPS each 10 ng/mL) for 2 h. The results were represented as mean ± SD of three independent experiments, * *p* < 0.05, ** *p* < 0.01, *** *p* < 0.001, **** *p* < 0.0001.

**Figure 5 cancers-14-02665-f005:**
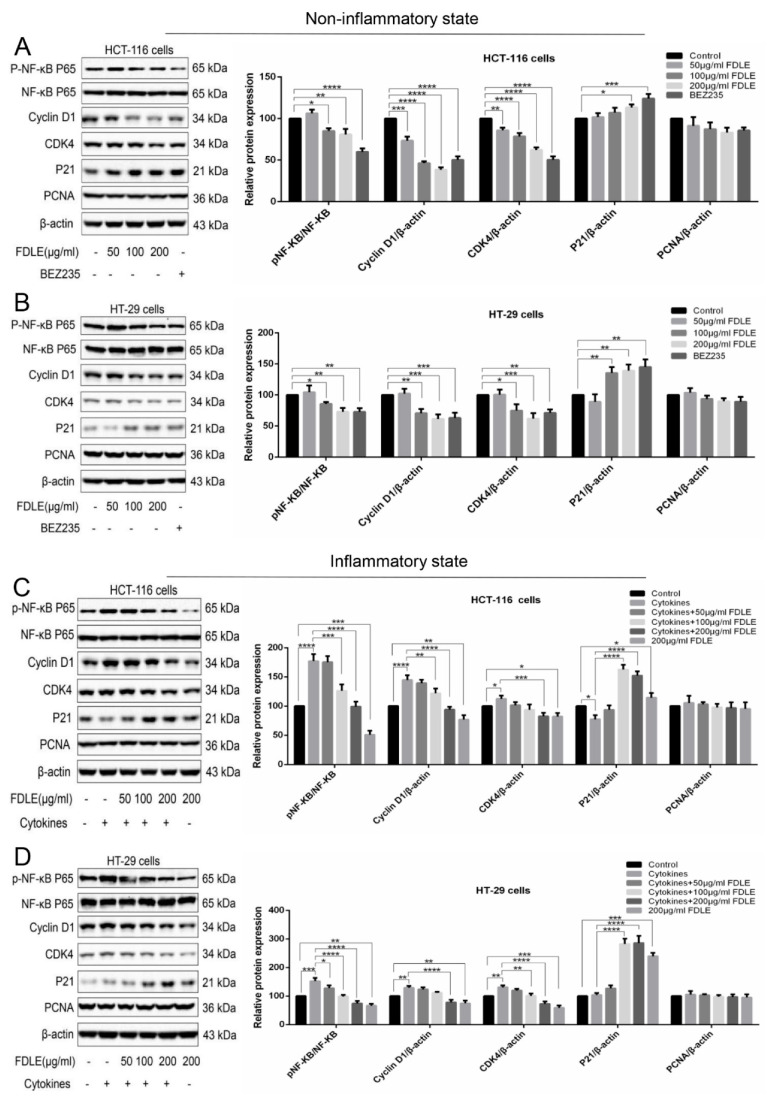
The effects of FDLE on NF-κB-mediated cell cycle proteins in non-inflammatory and inflammatory HCT-116 and HT-29 colorectal cancer cell line. (**A**,**B**) Western blot analysis of NF-κB-mediated proteins related to cell cycle in non-inflammatory HCT-116 and HT-29 cells treated with three doses (50, 100 and 200 µg/mL) of FDLE for 48 h. (**C**,**D**) Western blot analysis of NF-κB-mediated proteins related to cell cycle in inflammatory HCT-116 and HT-29 cells treated with three doses (50, 100 and 200 µg/mL) of FDLE for 48 h after adding a mixture of cytokines (TNF-α, IFN-γ and LPS each 10 ng/mL) for 2 h. Densitometric quantification was performed by image software. The results were represented as mean ± SD of three independent experiments, * *p* < 0.05, ** *p* < 0.01, *** *p* < 0.001, **** *p* < 0.0001. The whole western blots are shown in Appendix A.

**Figure 6 cancers-14-02665-f006:**
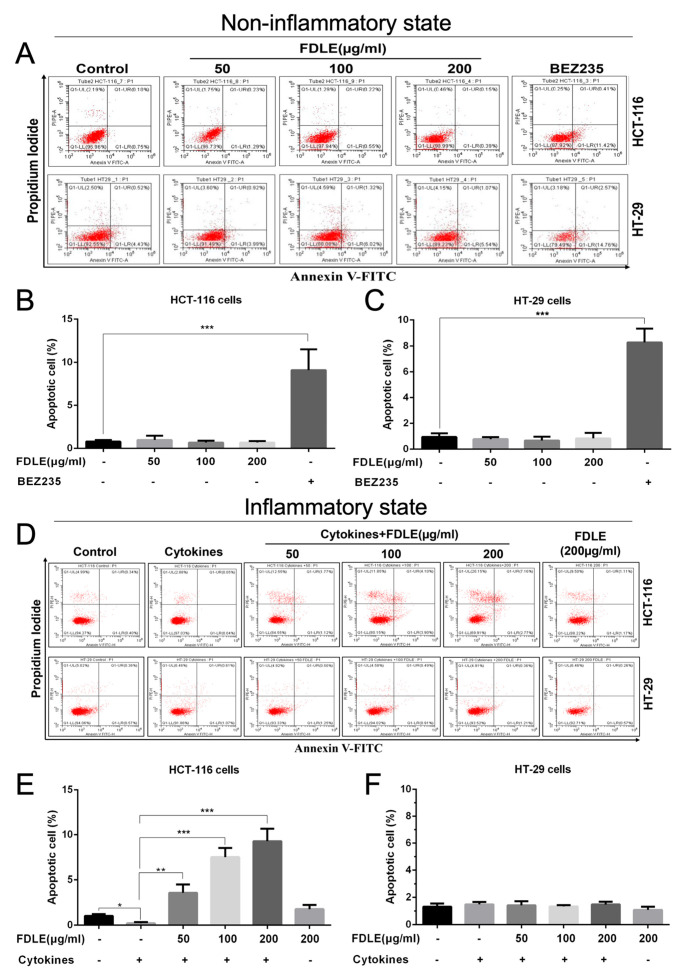
The effects of FDLE on apoptosis induction in non-inflammatory and inflammatory HCT-116 and HT-29 colorectal cancer cell lines. Apoptosis includes early apoptosis (LR) and late apoptosis (UR). (**A**–**C**) Apoptosis rate of non-inflammatory HCT-116 and HT-29 cells treated with three doses (50, 100 and 200 µg/mL) of FDLE for 48 h. (**D**–**F**) Apoptosis rate of inflammatory HCT-116 and HT-29 cells treated with three doses (50, 100 and 200 µg/mL) of FDLE for 48 h after adding a mixture of cytokines (TNF-α, IFN-γ and LPS each 10 ng/mL) for 2 h. The results were represented as mean ± SD of three independent experiments, * *p* < 0.05, ** *p* < 0.01, *** *p* < 0.0001.

**Figure 7 cancers-14-02665-f007:**
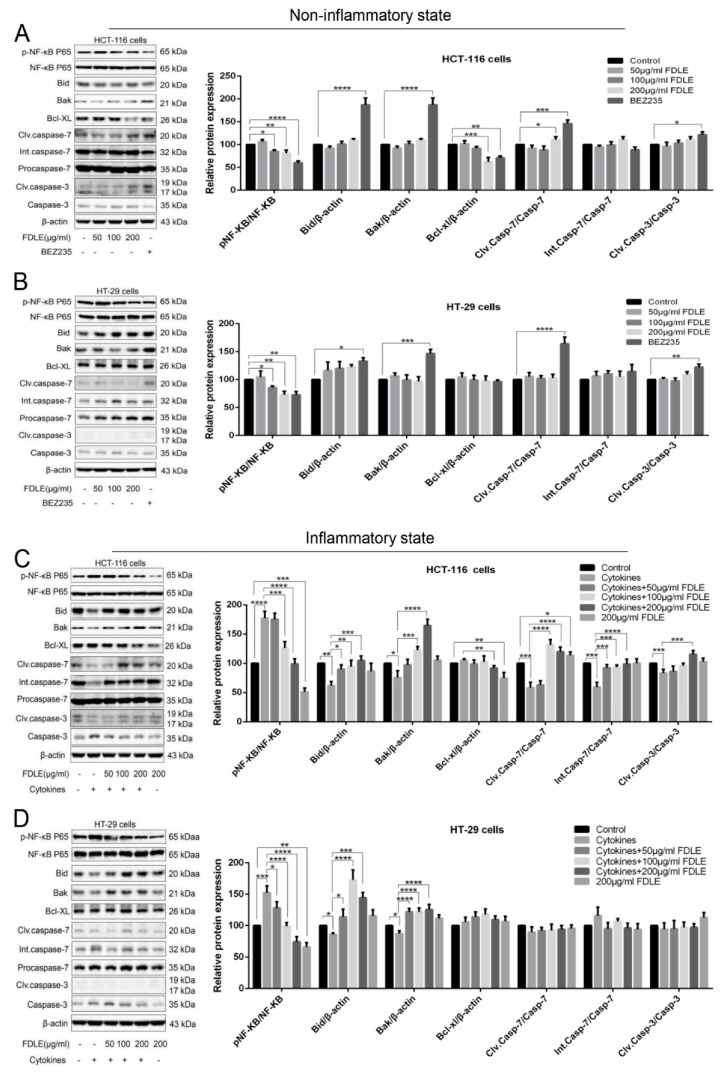
The effects of FDLE on NF-κB-mediated expression of apoptotic proteins and activation of caspase enzymes in non-inflammatory and inflammatory HCT-116 and HT-29 colorectal cancer cell lines by Western blot. (**A**,**B**) The expression of apoptotic proteins and the activation of caspase enzymes in non-inflammation HCT-116 and HT-29 cells untreated and treated with three doses (50, 100 and 200 µg/mL) of FDLE or BEZ235 for 48 h. (**C**,**D**) The expression of apoptotic proteins and the activation of caspase enzymes in inflammation HCT-116 and HT-29 cells untreated and treated with three doses (50, 100 and 200 µg/mL) of FDLE for 48 h after adding cytokines (TNF-α, IFN-γ and LPS each 10 ng/mL) for 2 h. Densitometric quantification was performed by image software. The results were represented as mean ± SD of three independent experiments, * *p* < 0.05, ** *p* < 0.01, *** *p* < 0.001, **** *p* < 0.0001. The whole western blots are shown in Appendix A.

**Figure 8 cancers-14-02665-f008:**
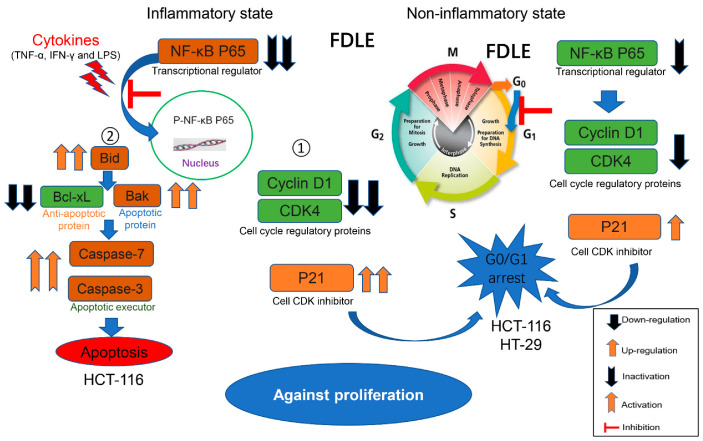
The different anti-proliferative mechanisms of FDLE on HCT-116 and HT-29 human colorectal cancer cell lines in non-inflammatory and inflammatory conditions.

## Data Availability

The data presented in this study are available on request from the corresponding author.

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
