# Peer review of "Ficus dubia Latex Extract Induces Cell Cycle Arrest and Apoptosis by Regulating the NF-κB Pathway in Inflammatory Human Colorectal Cancer Cell Lines"

_cancers, 2022, doi:10.3390/cancers14112665_

Round 1
Reviewer 1 Report
The authors studied the antineoplastic activity of Ficus dubia latex extract against human colorectal cancer cell lines (HCT- 116 and HT-29). Using MTT, flow cytometry and blots investigated proliferation, cell death, and inflammation.
The major concern is how an extract with an unknown concentration of compounds can be interpreted.
More information on the composition and stability of the compounds is required.
A better characterization of the inflammation model is also needed. It is not clear why using LPS and TNF-alpha together on those cell lines.
The MTT was the only experiment in which normal (non-tumor) cells were used to compare responses with tumor cell lines.
Minor points:
Please describe how IC was calculated.
Figure 1 the proliferating index of NHI cells was similar to cancer cell lines?
Figure 4: The flow cytometry histograms are too small, and difficult to see.
Figure 5: Please describe better in the text, the non-inflammatory condition, from which FDLE concentration was seen an increase and decrease of the cited proteins, for each cell.
Mislabeled Legend: "(A-C) Western blot analysis of NF-κB-mediated proteins related to cell cycle in non-inflammatory." It would be A-B.
"(D-F) Western blot analysis of NF-κB-mediated proteins related to cell cycle in inflammatory." It would be C-D.
Figure 6: Flow cytometry figures are unclear, " Interestingly, a mixture of cytokines significantly reduced (increased)the number of apoptotic cells only in HT-116 cells but not in HT-29 cells."
Figure 7: Please describe better the results.
Figure 8, which concludes the results, is not so easy to interpret visually, perhaps a brief summary in the legend of the figure itself would help.
Author Response
Point-by-point response to reviewer 1
We are very grateful to reviewers for your serious and responsible review to our manuscript and thank you very much for giving us valuable comments and suggestions. These are important guiding significance for improving this manuscript and also our future scientific research. We would like to response and answer point by point based on the editor's and reviewers' comments as follows.
Comments from Reviewer 1:
- The major concern is how an extract with an unknown concentration of compounds can be interpreted.
Response:
In a previous study, we discovered that the extract of Ficus dubia latex includes 34.2% phenolic and 7.3% flavonoids; specifically, the amount of chlorogenic acid, one form of phenolic, is approximately 2.1% using HPLC compared to the chlorogenic acid standard, with an extraction yield of 7.67%. Moreover, by HPLC chromatogram analysis, our result found that chlorogenic acid was the predominant phenolic compound in FDLE which was 21.1 ± 0.6 mg chlorogenic acid (CGA)/g FDLE1. Therefore, we concluded that the crude extract of F. dubia contained chlorogenic acid at 21.1 ± 0.6 mg per gram of extract. If comparing to an extraction yield of 7.67% 2, the chlorogenic acid accounted for approximately 161.84 mg per gram of latex. However, the study of each compound should be considered to further investigate.
- More information on the composition and stability of the compounds is required.
Response:
As mentioned, more information on the composition and stability of the compounds should be provided. From the previous study, Chansriniyom, C. et al. already investigated the chemical components of the Ficus dubia sap water extract by liquid chromatography-electrospray ionization quadrupole time-of-flight mass spectrometry (LC–ESI–MS/QTOF-MS) and found that the sap extract was composed of quinic acid and caffeoyl derivatives (syringoylquinic acid, 3-O-caffeoylquinic acid, 4-O-caffeoylquinic acid, and dimeric forms of caffeoylquinic acids)2. Despite the fact that the stability of FDLE has not been investigated, the pharmacokinetics of chlorogenic acids, the major component of FDLE, has been studied by various of researchers in humans and animals 3. Moreover, Ianni, F and Narita, Y indicated that the degradation of chlorogenic acid at 37 °C water and pH 7.4 to 9.0 satisfactorily correlated with the Weibull equation, the chemical environment or microwave processing may have an effect on the stability of chlorogenic acid in aqueous solution4,5. It is feasible that a substance that is soluble in water is stable based on this information. This information was partially added in Discussion part.
- A better characterization of the inflammation model is also needed. It is not clear why using LPS and TNF-alpha together on those cell lines.
Response:
Inflammatory mediators such as TNF-α, IL-6 and IL-1β are generally responsible for the presence of inflammation in the tumor microenvironment as well as in colorectal cancer6,7. Consequently, TNF-α caused inflammation in cancers was commonly implemented in numerous investigations8. However, there is no evidence on the use of TNF- α and LPS to induce cell proliferation and inflammation in cancer cells. According to a previous study, LPS could affect the expression of inflammatory factors such as tumor necrosis factor-α (TNF-α), interleukin-6 (IL-6), and cyclooxygenase-2 (COX-2) in the SW480 and HCT116 colorectal cancer cell lines via NF-kB signaling pathway, but had no effect on cell proliferation9,10. In order to induce inflammation and proliferation through the activation of NF-kB, we will conduct an experiment including the stimulation of both TNF-α and LPS, with the purpose of comparing the effect of FDLE on the proliferation of colorectal cancer cells under normal and inflammatory conditions and exploring its anti-proliferative mechanism under different conditions which is related to tumor microenvironment 11. We have added the description to Discussion.
(3) The MTT was the only experiment in which normal (non-tumor) cells were used to compare responses with tumor cell lines.
Response:
The purpose of our experiments was to evaluate whether FDLE could inhibit the proliferation of colorectal cancer cells and to explore the possible mechanism of anti-proliferation. We agree that the effect of FDLE on normal cells is crucial. In a previous study, it was suggested that FDS had no cytotoxic effect on the viability of HaCaT (human keratinocyte) cells at high concentrations (more than 200 g/ml)2. MTT is a standard method for detecting the cytotoxicity of medicines or prospective medicinal plant extracts in vitro. We hope that FDLE has an antiproliferative effect on tumor cells but not on normal cells. The quantities of FDLE utilized in our investigation did not significantly reduce the growth of normal cells (NIH3T3), indicating that it is not cytotoxic to normal cells with an IC50 greater than 1,000 g/ml (Supplementary table 1). We do apologize for provision of manuscript. However, the effect of FDLE should be further investigated into the possible mechanism on normal cells.
Moreover, in our animal experiment, FDLE extract did not show any signs of adverse effects; there was no substantial difference in body weight, organ weights between the control and treatment rats 1. That study suggests that FDLE is more sensitive to cancer cells than normal cells. We have added descriptions of the effect of FDLE on NIH3T3 cells in the first paragraph of Discussion.
- Please describe how IC was calculated.
Response:
Thank you very much for pointing this out. We analyzed the half-maximal inhibitory concentration (IC50) using the GraphPad in the mode of dose-response-inhibition, which is adapted for calculating the IC50. Many log(inhibitor) vs. response curves follow the familiar symmetrical sigmoidal shape. The goal is to determine the EC50 of the inhibitor-the concentration that provokes a response half-way between the maximal (Top) response and the maximally inhibited (Bottom) response 12.
The analysis process is as follows as shown in following figures, and the analyze other IC value, search the web: https://www.graphpad.com/quickcalcs/Ecanything1.cfm, then put the IC50 value and HillSlope value into the boxes respectively as shown in Fig 2G, and choose the desired EC according IC value what we want to calculate (for example EC80=IC20).
- Figure 1 the proliferating index of NIH cells was similar to cancer cell lines?
Response:
Under non-inflammatory conditions, our experiment revealed that FDLE exhibited strong anti-proliferative effects only in cancer cells but not normal cells (NIH3T3 cells). This finding suggests that only FDLE treatment is safe and effective for colorectal cancer cells in non-inflammatory conditions. Due to the anti-inflammatory properties of FDLE, this treatment was cytotoxic to both cancer and normal cells under inflammatory conditions. However, the IC50 for colorectal cancer cells treated with FDLE was about 200 g/ml, whereas the IC50 for colorectal cancer cells treated with FDLE was around 100 g/ml. (Supplementary data Table1). We agree that the cytotoxic effect on normal colonic epithelial cells is better used to confirm the safety of PDLE.
- Figure 4: The flow cytometry histograms are too small, and difficult to see.
Response:
We have modified these flow cytometry histograms. Because there are too many pictures, we compressed these pictures when we analyzed them. Thank you for your careful observation.
- Figure 5: Please describe better in the text, the non-inflammatory condition, from which FDLE concentration was seen an increase and decrease of the cited proteins, for each cell.
- Mislabeled Legend: "(A-C) Western blot analysis of NF-κB-mediated proteins related to cell cycle in non-inflammatory." It would be A-B.
- "(D-F) Western blot analysis of NF-κB-mediated proteins related to cell cycle in inflammatory." It would be C-D.
Response:
Thank you very much for your suggestion careful observation, we have re-posted with new Figures and describe better these Figures in the results section. We also modified the mislabeled Legend.
- Figure 6: Flow cytometry figures are unclear, " Interestingly, a mixture of cytokines significantly reduced (increased) the number of apoptotic cells only in HCT-116 cells but not in HT-29 cells."
Response:
Thank you very much for this correction point, we have re-posted with new Figure 6. In our experiment. We found that a mixture of cytokines significantly reduced the number of apoptotic cells only in HCT-116 cells but not in HT-29 in inflammation model. On the one hand, we think the number of apoptotic cells is very low in normal growth cancer cell. Although a mixture of cytokines can inhibit cell apoptosis, the difference is not particularly strong. Our experiment also indicated that the overall numbers of apoptosis in cytokines treatment group and control group were not strong in HCT-116 cell. Moreover, these two colorectal cancers cells used in the study had different mutations status in several critical genes involved in colorectal cancer in addition to BRAF (HCT-116 wt; HT-29 V600E) including P53 (HCT-116 wt; HT-29 mutation R273H) different type of gene mutations may lead to differential sensitivity to inflammatory or drug stimulation13. According to p53, p21 in tumor cell response to cytotoxic medicines, as well as the potential to increase the therapeutic index of such agents by altering p21 status through p53-dependent pathways, which was detected in HCT-116 14 but not in HT29, which has a deficiency in p53 gene expression 15. As a result, HCT-116 maybe be more sensitive to inflammation stimulus than HT-29. We have edited this part in the Discussion.
- Figure 7: Please describe better the results.
Response:
We have edited the explanation of Figure 7 in the results section. Thank you very much.
- Figure 8, which concludes the results, is not so easy to interpret visually, perhaps a brief summary in the legend of the figure itself would help.
Response:
We have made reasonable modifications of Figure 8 and carefully described in the conclusion section, thank you for your suggestion.
References
1 Hu, R. et al. Ficus Dubia Latex Extract Prevent DMH-Induced Rat Colorectal Carcinogenesis Through the Regulation of Xenobiotic Metabolism, Inflammation, Cell Proliferation and Apoptosis. (in press).
2 Chansriniyom, C. et al. Tandem mass spectrometry of aqueous extract from Ficus dubia sap and its cell-based assessments for use as a skin antioxidant. Scientific reports 11, 1-13 (2021).
3 Xie, X., He, X., Qiu, X. & Song, Z. In vitro monitoring chlorogenic acid in human urine and serum by a flow injection system exploiting the luminol-dissolved oxygen chemiluminescence reaction. Current drug metabolism 8, 773-777 (2007).
4 Ianni, F. et al. Investigation on chlorogenic acid stability in aqueous solution after microwave treatment. Food Chemistry 374, 131820 (2022).
5 Narita, Y. & Inouye, K. Degradation kinetics of chlorogenic acid at various pH values and effects of ascorbic acid and epigallocatechin gallate on its stability under alkaline conditions. Journal of Agricultural and Food Chemistry 61, 966-972 (2013).
6 Silva, A. L., Faria, M. & Matos, P. Inflammatory microenvironment modulation of alternative splicing in cancer: a way to adapt. Tumor Microenvironment, 243-258 (2020).
7 Long, A. G., Lundsmith, E. T. & Hamilton, K. E. Inflammation and colorectal cancer. Current colorectal cancer reports 13, 341-351 (2017).
8 Yang, M. et al. Astragalin Inhibits the Proliferation and Migration of Human Colon Cancer HCT116 Cells by Regulating the NF-κB Signaling Pathway. Frontiers in Pharmacology 12, doi:10.3389/fphar.2021.639256 (2021).
9 Cha, J. H., Kim, W. K., Ha, A. W., Kim, M. H. & Chang, M. J. Anti-inflammatory effect of lycopene in SW480 human colorectal cancer cells. Nutrition Research and Practice 11, 90-96 (2017).
10 ZHU, G. et al. TRAF6-Mediated Inflammatory Cytokines Secretion in LPS-induced Colorectal Cancer Cells Is Regulated by miR-140. Cancer Genomics - Proteomics 17, 23-33, doi:10.21873/cgp.20164 (2020).
11 Schottelius, A. J. & Dinter, H. Cytokines, NF-κB, microenvironment, intestinal inflammation and cancer. The Link Between Inflammation and Cancer, 67-87 (2006).
12 Motulsky, H. J. Prism 5 statistics guide. GraphPad Software Inc.: San Diego, CA, USA (2007).
13 Berg, K. C. et al. Multi-omics of 34 colorectal cancer cell lines-a resource for biomedical studies. Molecular cancer 16, 1-16 (2017).
14 Ravizza, R., Gariboldi, M. B., Passarelli, L. & Monti, E. Role of the p53/p21 system in the response of human colon carcinoma cells to Doxorubicin. BMC Cancer 4, 92, doi:10.1186/1471-2407-4-92 (2004).
15 Davidson, D. et al. Irinotecan and DNA-PKcs inhibitors synergize in killing of colon cancer cells. Investigational new drugs 30, 1248-1256 (2012).
16 Ridgway, P. Revised fixed dose procedure: OECD Test Guideline 420. RAPPORTI ISTISAN, 27-31 (2002).

Reviewer 2 Report
This work by Rentong Hu et all aims to examine the anticancer activity of Ficus dubia lactox extract in HTC-116 and HT-29 cell lines. Ficus dubia induced cell cycle arrest by decreasing the expression of NF-kB, Cyclin D1, CDK4 and by increasing the expression of p21 in non-inflammatory and inflammatory HCT-116 and HT29 cells.
My minor comments are:
- I suggest extending the introduction paragraph;
- review figure 5 because the legend of the graphs is hard to read
- in figure 6 panel B-C why is there no significance in the three doses (50, 100 and 200 µg / mL) of FDLE for 48 h?
Author Response
Point-by-point response to reviewer 2
We are very grateful to reviewers for your serious and responsible review to our manuscript and thank you very much for giving us valuable comments and suggestions. These are important guiding significance for improving this manuscript and also our future scientific research. We would like to response and answer point by point based on the editor's and reviewers' comments as follows.
Response to reviewer 2:
- I suggest extending the introduction paragraph;
Response:
Thank you for your suggestion, we already extended the introduction paragraph by adding more information about rationale and focus.
- Review figure 5, because the legend of the graphs is hard to read.
Response:
Thank you very much for your suggestion and careful observation, we have re-posted with new Figure and describe better this Figure in the results section. We have already modified the mislabeled Legend.
- in figure 6 panel B-C why is there no significance in the three doses (50, 100 and 200 µg / mL) of FDLE for 48 h?
Response:
B and C are the result of apoptosis under non-inflammatory conditions, which showed that the FDLE used for our experiment is low toxicity or non-toxic mixture. In addition, the previous work has exhibited that no side effects or toxicity in vivo at a dose of more than 5,000 mg/kg body weight by acute toxicity test according to OECD test Guideline 42016. In vitro experiment also found that the values of IC50 were both higher than 300 µg/ml (Supplementary Table S1) when FDLE treatment for 48 hours in both non-inflammatory HCT-116 and HT-29 cells, which indicated the FDLE has not cytotoxicity17,18. On the other hand, the extract showed slightly inhibition of cell proliferation by inducing cell cycle arrest. Similarly, Tamura, K. et al. also found that Angiomodulin (AGM) treatment suppresses VEGF-A-induced tube formation, cell migration and proliferation, but does not induce apoptosis19,20. In our experiment, the highest concentration of FDLE (200 µg/mL) we use did not induce apoptosis in non-inflammatory conditions, therefore, there no significance in the three doses (50, 100 and 200 µg / mL) of FDLE for 48 h.
References
16 Ridgway, P. Revised fixed dose procedure: OECD Test Guideline 420. RAPPORTI ISTISAN, 27-31 (2002).
17 Weerapreeyakul, N., Nonpunya, A., Barusrux, S., Thitimetharoch, T. & Sripanidkulchai, B. Evaluation of the anticancer potential of six herbs against a hepatoma cell line. Chinese medicine 7, 1-7 (2012).
18 Indrayanto, G., Putra, G. S. & Suhud, F. Validation of in-vitro bioassay methods: Application in herbal drug research. Profiles of Drug Substances, Excipients and Related Methodology 46, 273-307 (2021).
19 Tamura, K., Yoshie, M., Hashimoto, K. & Tachikawa, E. Inhibitory effect of insulin-like growth factor-binding protein-7 (IGFBP7) on in vitro angiogenesis of vascular endothelial cells in the rat corpus luteum. Journal of Reproduction and Development (2014).
20 Tamura, K. et al. Insulin-like growth factor binding protein-7 (IGFBP7) blocks vascular endothelial cell growth factor (VEGF)-induced angiogenesis in human vascular endothelial cells. European journal of pharmacology 610, 61-67 (2009).

Reviewer 3 Report
Colorectal cancer (CRC) is the second most deadly cancer. Exploring the molecular mechanisms of drug action is crucial to understanding the effectiveness of drugs as well as safety. In this study, Hu et al. explored the possible different mechanisms in non-inflammatory and inflammatory states of CRC cell lines using the Ficus dubia latex extract. Overall, the manuscript is well organized and written. However, there are still some issues that need to be solved before publishing.
The summary needs to be improved. It seems that it is another version of the abstract. Authors can write the key findings and precise conclusions based on the results. Use the full form of the abbreviation at the first appearance in the text. Follow this throughout the manuscript.
The objective of this study is not well written in the abstract. It must be aligned with the introduction.
The introduction must be improved. Authors can start with the current status of CRC. What is the treatment status? Why is plant-based treatment important for CRC management? Authors suddenly brought the Ficus dubia into the contents. Having coherence in the content is obligatory. Write the problem statement clearly and how this study fulfils the existing problem is also part of the introduction. Write the complete botanical of the used plant. Authors may find the complete name here http://www.theplantlist.org/. Always italicized scientific names throughout the manuscript.
The methodology is sufficiently written. However, a comparison of inflammatory and non-inflammatory CRC cell lines is absent. This needs to be included in the results and figures as well.
The comparison of the treatment groups and control groups is not sufficiently discussed. Comparing the results with other studies of Ficus dubia extract against an antiproliferative activity as well as plants with the same genus is important but it is lacking in the discussion. As this study is on cell lines, compare the results with the cell lines study, not in vivo. However, in vivo studies can be brought to support the cell lines findings.
The conclusion does not support the results as the authors did not compare inflammatory and non-inflammatory CRC cell lines statistically.
Author Response
Point by point response to reviewer 3
We are very grateful to reviewers for your serious and responsible review to our manuscript and thank you very much for giving us valuable comments and suggestions. These are important guiding significance for improving this manuscript and also our future scientific research. We would like to response and answer point by point based on the editor's and reviewers' comments as follows.
Response to reviewer 3:
- The summary needs to be improved. It seems that it is another version of the abstract. Authors can write the key findings and precise conclusions based on the results. Use the full form of the abbreviation at the first appearance in the text. Follow this throughout the manuscript.
Response:
We already modified the summary and added the full form of the abbreviation at the first appearance in the manuscript according to you suggestion, thank you very much for your kindly comments.
- The objective of this study is not well written in the abstract. It must be aligned with the introduction.
The introduction must be improved. Authors can start with the current status of CRC. What is the treatment status? Why is plant-based treatment important for CRC management? Authors suddenly brought the Ficus dubia into the contents. Having coherence in the content is obligatory. Write the problem statement clearly and how this study fulfils the existing problem is also part of the introduction. Write the complete botanical of the used plant. Authors may find the complete name here http://www.theplantlist.org/. Always italicized scientific names throughout the manuscript.
Response:
We have edited the abstract and introduction part based on your suggestion.
- The methodology is sufficiently written. However, a comparison of inflammatory and non-inflammatory CRC cell lines is absent. This needs to be included in the results and figures as well.
Response:
We have described the differences between non-inflamed and inflammatory model in the Methods section. Since both the non-inflammatory model and the inflammatory model use the same colorectal cancer cell lines, we did not compare the inflammatory and non-inflammatory CRC cell lines. But in the result section, we already compared the different results of the same cell line under non-inflammatory and inflammatory conditions (Supplementary Table S1-6). Meanwhile, we explain the possible reasons for these differences in the discussion section. We think that an inflammation-activated NF-κB increased proliferation-related proteins and decreased apoptosis-related proteins, leading to hyperproliferation of colorectal cancer cell lines. FDLE exhibited effectively anti-proliferation through its ability to regulate proteins related to cell proliferation and apoptosis in inflammatory conditions, which may be due to its anti-inflammatory activity.
- The comparison of the treatment groups and control groups is not sufficiently discussed. Comparing the results with other studies of Ficus dubia extract against an antiproliferative activity as well as plants with the same genus is important but it is lacking in the discussion. As this study is on cell lines, compare the results with the cell lines study, not in vivo. However, in vivo studies can be brought to support the cell lines findings.
Response:
In the non-inflammatory model, we compared the difference between the treatment group and the control group. However, in the inflammatory model, we also compared the difference between the inflammation and the control group, but we paid more attention to the changes of cell proliferation and apoptosis after FDLE treatment in the presence of inflammation. Therefore, we mainly compared the difference between the treatment group and the inflammation group. Then, we use the same treatment. we hope to compare the anti-tumor activity of Ficus dubia latex extract (FDLE) against HCT-116 and HT-29 human colorectal cancer cell lines in normal and inflammatory condition and explore its mechanism of action. About the comparison of Ficus dubia and other Ficus species, since there are few articles about Ficus dubia, especially the anti-proliferative activity in vitro, we can only choose other types of Ficua to compare, we have referenced some related articles in the introduction section and the discussion section, these articles are mainly about anti-proliferative studies under normal growth conditions and anti-inflammatory in vivo, there are very few anti-proliferative studies under inflammatory conditions in vitro.
- The conclusion does not support the results as the authors did not compare inflammatory and non-inflammatory CRC cell lines statistically.
Response:
We compared inflammatory and non-inflammatory CRC cell lines statistically in every experiment (Supplementary Table S1-6) and explained the possible mechanism of this difference in the discussion section. Moreover, we expressed the parameters as relative value to control treatment of each group, therefore it was hard to statistically compare the effective of FDLE against colorectal cancer cell lines between normal and inflammatory condition. However, we tried to modify Figure 8 and edit the conclusion which supported to the results. Thank you very much for your suggestion.

Round 2
Reviewer 1 Report
Dear authors,
The manuscript has improved, but the overall interest is limited since it is used poorly characterized extracted.